# Growth Performance and Adaptability of European Sea Bass (*Dicentrarchus labrax*) Gut Microbiota to Alternative Diets Free of Fish Products

**DOI:** 10.3390/microorganisms8091346

**Published:** 2020-09-03

**Authors:** David Pérez-Pascual, Jordi Estellé, Gilbert Dutto, Charles Rodde, Jean-François Bernardet, Yann Marchand, Eric Duchaud, Cyrille Przybyla, Jean-Marc Ghigo

**Affiliations:** 1Unité de Génétique des Biofilms, Institut Pasteur, UMR CNRS2001, 75015 Paris, France; david.perezpascual@pasteur.fr; 2AgroParisTech, GABI, Université Paris-Saclay, INRAE, 78350 Jouy-en-Josas, France; jordi.estelle@inrae.fr; 3Laboratoire Service d’Expérimentations Aquacoles, Ifremer, 34250 Palavas les flots, France; gilbert.dutto@ifremer.fr; 4MARBEC, IRD, IFREMER, Univ Montpellier, CNRS, Laboratoire Adaptation, adaptabilité, des animaux et des systèmes Ifremer, 34250 Palavas les flots, France; charles.rodde@ifremer.fr; 5Virologie et Immunologie Moléculaires, INRAE, Université Paris-Saclay, 78350 Jouy-en-Josas, France; jean-francois.bernardet@inrae.fr (J.-F.B.); eric.duchaud@inrae.fr (E.D.); 6Le Gouessant, F-22402 Lamballe, France; yann.marchand@legouessant.fr

**Keywords:** European sea bass, fish gut microbiota, bacteria

## Abstract

Innovative fish diets made of terrestrial plants supplemented with sustainable protein sources free of fish-derived proteins could contribute to reducing the environmental impact of the farmed fish industry. However, such alternative diets may influence fish gut microbial community, health, and, ultimately, growth performance. Here, we developed five fish feed formulas composed of terrestrial plant-based nutrients, in which fish-derived proteins were substituted with sustainable protein sources, including insect larvae, cyanobacteria, yeast, or recycled processed poultry protein. We then analyzed the growth performance of European sea bass (*Dicentrarchus labrax* L.) and the evolution of gut microbiota of fish fed the five formulations. We showed that replacement of 15% protein of a vegetal formulation by insect or yeast proteins led to a significantly higher fish growth performance and feed intake when compared with the full vegetal formulation, with feed conversion ratio similar to a commercial diet. 16S rRNA gene sequencing monitoring of the sea bass gut microbial community showed a predominance of *Proteobacteria*, *Firmicutes*, *Actinobacteria,* and *Bacteroidetes* phyla. The partial replacement of protein source in fish diets was not associated with significant differences on gut microbial richness. Overall, our study highlights the adaptability of European sea bass gut microbiota composition to changes in fish diet and identifies promising alternative protein sources for sustainable aquafeeds with terrestrial vegetal complements.

## 1. Introduction

Aquaculture is the fastest growing animal-food producing sector, supplying over half of fish and seafood for human consumption [1]. Among the nutrient constituents of aquaculture feeds, animal protein sources are the most expensive ingredients, as fish feeds still contains significant levels of fishmeal [1,2]. With the reduction of available oceanic resources, alternatives to fishmeal (FM) which maintain nutritional quality for farmed fish and human consumption need to be identified to ensure sustainability and reduce the environmental impact of the farmed fish industry [3,4,5,6]. While terrestrial plants, marine algae, yeast, and alternative animal proteins, including insects, are currently considered as new protein sources, their impact on fish gut microbiota, intestine health, and growth performance are not yet fully known [7,8,9,10].

Farmed European sea bass (*Dicentrarchus labrax* L.) is of key economic importance in Europe, since 96% of the total production originates from aquaculture rather than from fisheries [11]. Previous studies have shown that partially substituting fishmeal by terrestrial plant protein extract alone or in combination with vegetal oils does not affect European sea bass survival and growth performance [4,12,13,14,15,16,17]. However, the use of vegetal proteins as an alternative source is associated with several disadvantages, including low palatability and digestibility [18], lack of essential omega−3 fatty acids (EPA and DHA) [19] and presence of many anti-nutritional factors [20]. Proteins from poultry, brewer’s yeast (*Saccharomyces cerevisiae*), larvae insect meal (*Hermetia illucens* and *Tenebrio molitor*), or microalgae were reported to advantageously replace up to 76% of fishmeal without significantly impacting European sea bass performance [21,22,23,24,25,26]. However, data concerning the effects of fishmeal substitution on sea bass gut microbiota are scarce, and only focused on partial replacement of fishmeal by alternative protein sources [4,27].

Fish intestinal microbiota plays a major role in host health and physiology [28], promoting the development of the immune system and contributing to nutriment utilization and resistance against pathogens [29,30]. Fish gut microbial communities are also highly dynamic and respond rapidly to variations in local selective pressures such as diet modification [31,32,33]. While such plasticity contributes to host adaptation to environmental changes [33,34], it can also lead to microbiota imbalances that could negatively affect fish growth and disease sensitivity [35]. Hence, the impact of reduction of fishmeal and fish oil by introducing new ingredients in fish farming diets on fish gut microbiota composition and diversity needs to be carefully evaluated [10,36,37].

Here, we investigated how European sea bass growth performance and gut microbiota composition are affected by diets exclusively composed of terrestrial photosynthetic sources with 15% replacement by alternative proteins originating either from insect, terrestrial animal by-products, cyanobacteria, or yeast. We showed that insects and yeast constitute promising protein complements to vegetal fish feeds. This study of the relationships between change in fish diet, gut microbiota composition, and fish growth provides insights into how to improve fish farming sustainability.

## 2. Materials and Methods

### 2.1. Ethical Statement

All animal experiments described in the present study were conducted at the Ifremer aquaculture research station of Palavas-les-Flots in accordance with the recommendations of Directive 2010−63-EU on the protection of animals used for scientific purposes. The protocols were approved by C2EA−36 (“Comité d’éthique en expérimentation animale Languedoc–Roussillon”) under authorization APAFiS n°# 9006- 2017011611369885, approved on 06/15/2019.

### 2.2. Feed Formulation and Feeding

Specific formulations for sea bass were calculated by Le Gouessant Aquaculture^®^ company and produced in small scale by INRAE-Donzacq, using the same machine tools. Five formulations were processed without fishmeal (FM) and fish oil (FO) content (Figure 1). One of these was composed exclusively of terrestrial photosynthetic sources (VEG) using wheat gluten, soy concentrate, pea protein, and rapeseed oil. The VEG condition was considered as a control condition. The four other feed formulas were composed of approximately 85% VEG and 15% experimental alternatives proteins sources: cyanobacterium (*Arthrospira platensis*, SPI), processed animal protein composed by a mix of poultry blood meal and hydrolyzed poultry feathers (PAP), yeast protein fraction from *Saccharomyces cerevisiae* (YEA) (Nutrisaf^®^ 503, Phileo by Lesaffre, France), and an hydrolyzed insect larvae meal (*Hermetia illucens*) (INS) (Copalis, France). An additional control diet was formulated with fishmeal and fish oil similar to conventional commercial formulations for European sea bass (COM). All diets, except COM, were supplemented with DHA-rich marine microalgae (*Schizochytrium* sp) biomass to supply sufficient n−3 highly unsaturated fatty acids to the fish (Appendix A). Formulations were applied according to sea bass requirements in fatty acids and essential amino acids [38]. Experimental diets were isoproteic (42.6 ± 0.6%), and isoenergetic (21.8 ± 0.3% kJ.g^−1^).

### 2.3. Fish and Experimental Conditions

Four thousand sea bass fingerlings were produced at Ifremer laboratory L−3AS (Palavas-les Flots, France). After a selection, 848 fish (19.32 ± 1.6 g) were tagged and randomly assigned to sixteen one cubic meter tanks (53 fish/tank) in a recirculating aquaculture system (RAS) (Appendix A) [39]. Tank temperature was adjusted at 23.2 ± 0.7 °C all over the experiment. Salinity was recorded at 37 ± 0.1 PSU (practical salinity unit) and pH was maintained at 8.13 (±0.08). Dissolved oxygen concentration in rearing water was maintained at 90% saturation in the tank outlet for an optimal sea bass growth and welfare [40]. Photoperiod was 16 h of light per day and the intensity at the center of the tank was around 316 lux (±27 lux).

Fish were acclimatized with a commercial diet for one month, when all groups showed a feeding behavior consistent with the stocking density and development stage [39,41]. Three tanks were used for each experimental formulation (VEG-control, SPI, YEA, INS, and PAP), and one tank was used for the commercial condition COM. The COM tank was not considered for statistical analysis of growth performance but used as an informative growth curve baseline. In classical fish nutrition experiments, the diet is evaluated until the fish body weight has tripled [41]. This single COM tank was our biological indicator to decide at which time to stop the experiment and the source of control fish gut samples for analyzing bacterial biodiversity. Fish were fed manually twice a day. Uneaten pellets were trapped in a central drain located at the bottom of the tank. The daily feed ingested per tank was calculated using the following formula: F_Ing_ = F _distributed_ − F_trapped,_ (the average weight of one pellet was known for each formulation and the variability of pellets weight was low within each formulation). All fish (*n* = 848) were weighed at the start of the experiment (day 0) and at days 28, 62, and 93, corresponding to the end of the experiment. The initial biomass (B_i_) and final biomass (B_f_) were normalized to the survival ratio. Survival (%) = 100 × (N_f_/N_i_), where N_i_ and N_f_ are the initial and final fish number respectively. The feed conversion ratio was calculated using the following formula: FCR = ΣF_Ing/_(Bf − B_i_).

### 2.4. Digestibility Methods

Yttrium oxide (0.01%) was incorporated into the experimental diets during the feed manufacture to calculate the digestibility rate for the main compounds. For each tank, the feces emitted by the fish were drawn into a central drain at the bottom of the tanks. Particles were stocked into the fecal trap for 15 h after feeding. The fish feces from each tank were collected daily during the last three weeks of the feeding trial and preserved at −20 °C. Fecal biomass was freeze-dried and both feed and feces were analyzed for dry matter, protein, and lipids. Yttrium oxide concentration in feed and feces was determined by inductively coupled plasma mass spectrometry (ICP-MS). The apparent digestibility coefficients (ADCs) of the dry matter, protein, lipid and energy of the diets were calculated as follows: ADC = $1\;–\;\left[\left(\frac{F}{D}\right)\;*\;\left(\frac{Di}{Fi}\right)\right]$, where: D = % for the nutrient or KJ/g for the energy in the diet; F = % for the nutrient or KJ/g for the energy in the feces; Di = % yttrium oxide in the diet; and Fi = % yttrium oxide in the feces.

### 2.5. Sampling of Gastrointestinal Contents

Sampling of intestinal content to characterize gut bacterial community was performed at the end of the trial on day 93, after approximately 44 h of fasting. Five fish per tank were euthanized using cold seawater (around 0 °C). The whole intestine, from just below the pyloric caeca to immediately before the anus, was dissected under sterile conditions and the intestinal content was squeezed out individually into an Eppendorf tube and placed at −80 °C until DNA extraction [42]. A total of 131 intestinal content samples were collected. In addition, four 2-mL samples of input water as well as 250 mg samples of each experimental feed were collected for analysis of their bacterial content.

### 2.6. DNA Extraction and 16S rRNA Gene Sequencing and Analyses

Total bacterial DNA was extracted from each gut content sample using the QIAmp DNA Microbiome kit (Qiagen), following the manufacturer’s instructions. DNA concentrations were measured using a Nanodrop™ 1000 (Thermo Scientific Ltd.). All reagents used were molecular grade and supplied by Sigma-Aldrich (UK). Samples with DNA concentration >10 ng/μL with an absorbance ratio A260:A280 of >1.8 were kept for further analysis. 16S rRNA gene amplification, library construction and sample sequencing were performed by PCR of the V3-V4 region with 10 ng of DNA and 200 nM of the degenerated primers F_NXT_N341F (5′-cctacgggrsgcagcag-3′) and R_NXT_BAKT_805R and (5′-actachvgggtatctaatcc-3′) [43]. PCR products were purified using solid phase reversible immobilization (SPRI) paramagnetic bead-based technology (AMPure XP beads, Beckman Coulter) with a bead:DNA ratio of 0.7:1 (v/v). and quantified using a Qubit^®^ 2.0 Fluorometer (Invitrogen). Sequencing was performed at a final concentration of 4 pM with a 10% PhiX control library spike-in on a MiSeq instrument (Illumina Inc.) with 500 cycle v2 chemistry to generate 2 × 250 bp paired-end reads.

### 2.7. Bioinformatic and Biostatistical Analyses of Microbiota Datasets

Raw sequences were deposited in Sequence Read Archive (SRA) OF NCBI under SUB7839880 accession number. The quality of the FastQ files were first validated with the FastQC software (https://www.bioinformatics.babraham.ac.uk/projects/fastqc/) and then analyzed using the Quantitative Insights Into Microbial Ecology (QIIME) package v1.9.1 [44] with the open-reference sub-sampled OTU calling strategy [45] following the authors’ recommendations and using the GreenGenes 13_8 reference database [46]. Singleton OTUs and OTUs less than 0.005% of the total number of sequences per sample were removed from the dataset as recommended [47]. Chimeric sequences were removed using QIIME and using the BLAST algorithm. The biom OTU table was imported into R (v3.4.4) with Phyloseq v1.22.3 package [48]. Vegan v2.5−3 software package [49] was used for the rarefaction on the OTU level of each experimental group in R. Richness and diversity analyses were performed at the OTU level. Alpha diversity was calculated with Shannon index and richness was evaluated as the total number of OTUs present in each sample. Beta diversity was calculated with Whittaker’s index. ANOVAs were performed on alpha and beta diversity and changes in individual families were evaluated with log10 (richness) using the “aov” procedure in R. Vegan’s non-metric multidimensional scaling (NMDS), using the Bray–Curtis distance and with the “metaMDS” function that standardizes the scaling in the result, was used to represent the global diversity of gut microbiota composition between sample groups. The function “env_fit” in Vegan was used to fit environmental factors onto the NMDS ordination to compare the groups and evaluate the statistical significance. The permutational multivariate analysis of variance using distance matrices was performed using the “adonis” function in Vegan. The significance threshold was chosen at *p* < 0.05. The differential abundance analysis of phyla and genera were performed using the function fitZig in the metagenomeSeq v1.26.3 package [50].

### 2.8. Statistical Analyses of Body Weight Measures

Fish performance was statistically analyzed for the various diets except COM. Changes in fish body weight over time was studied with the following ANCOVA model:W_ijk_ = μ + D_i_ + R_j(i)_ + T_k_ + I_ik_ + ε_ijk_,(1)
where W_ijk_ is the weight of fish fed with diet i, reared in tank j, and measured at time step k (k between 1 and 4), μ is the general mean, D_i_ is the fixed effect of diet i, R_j(i)_ is the random effect of tank j that is nested into diet i with R_j(i)_ ~ N(0; σ^2^_r_), The Tk term is the fixed effect of time step k, I_ik_ is the interaction effect of diet i by time step k, and ε_ijk_ the residual (ε_ijk_ ∼ N(0;σ^2^_e_)). The following ANOVA model was used to study feed intake, feed conversion ratio and digestibility:F_i_ = μ + D_i_ + ε_i_,(2)
where F_i_ is the measured performance (feed intake, feed conversion ratio or digestibility) of fish fed with diet i, μ is the general mean, D_i_ is the fixed effect of diet i and ε_i_ the residual (ε_i_ ∼ N(0;σ^2^_e_)). The normality of residuals was checked using the quantile-quantile method (comparing residuals quantiles with theorical normal quantiles), and their homoscedasticity and independence by comparing residuals with the model fitted values. Linear mixed models and Student tests associated to these models were generated using R packages ”lme4” v1.1–20 [51] and “lmerTest” v3.1−0 [52]. Finally, Tukey tests were used to rank diets in terms of final body weight, feed intake and feed conversion ratio using R package «lsmeans» v2.30−0 [53].

## 3. Results and Discussion

### 3.1. Changing Protein Sources in Sea Bass Diet Influences Growth Performance and Feed Efficiency

To evaluate the impact of protein diets on fish growth performance, only fish with an initial weight of 19.32 ± 1.6 g were used to stock experimental tanks. Control fish were fed a terrestrial plant-based diet (VEG) while other groups were fed diets with 15% of their plant protein source replaced by the following: insect larvae (INS), transformed poultry proteins (PAP), cyanobacteria (*Spirulina platensis,* SPI), or yeast (YEA). In parallel, fish from the same batch were fed with conventional commercial diet (COM) (Figure 1). After 93 days, fish fed with the COM diet had tripled their weight (60.1 ± 2.0 g), thus meeting the experimental duration requirement to compare growth among diets. Analysis of body weight gain for each condition (*n* = 159/diet) revealed large intra-group individual variability increasing with time within all feed conditions, with a significant impact of time and diet on fish performance (*p* < 0.001, ANCOVA) (Figure 2). INS and YEA final body weight was significantly higher than PAP (*p* < 0.01), itself higher than VEG (*p* < 0.001), in turn significantly higher than SPI, which had the lowest body weight (*p* < 0.01). Diets exclusively composed of photosynthetic matter (VEG and SPI diets) led to lower growth performance, with even some negative weight slopes before recovery of a growth trend similar to other diets after day 28 (Figure 2, *p* < 0.001). In all diet conditions, fish survival rates were >97.9%, indicating no negative impact on overall fish survival.

Moreover, feed conversion ratios (FCR) differed between the diets (*p* < 0.001, ANOVA), with the YEA and INS diets showing the best ability to invest energy for growth, as indicated by their significantly lower FCR values (Figure 3A). Similar to FCR, feed intake (FI) was also impacted by the diet (*p* < 0.001, ANOVA). Fish fed YEA, INS, and PAP exhibited the highest feed intakes, while those fed VEG and SPI had the lowest feed intakes (Figure 3B). Such discrepancies in feed intake may be due to differences in pellets color, hardness, density (e.g., about one third of SPI pellets were floating) or permeability to water among the diets, that could impact fish feeding behavior. Since it has been demonstrated that fish can perceive flavors, even from molecules dissolved in water [54], it is also possible that feed intake varies with the raw materials included in the different feeds.

The differences observed in both feed intake (FI) and feed conversion ratio (FCR) explain differences in weight, with fish fed INS and YEA exhibiting the highest weight increase and those fed SPI being the smallest. Indeed, stronger growth with INS and YEA resulted from both higher FI and improved FCR. In contrast, SPI and VEG feed intake were the lowest, but SPI had an FCR significantly higher than that of VEG. The degraded FCR associated to the SPI diet may be explained by a low feed intake, previously shown to result in decreased FCR [55,56]. This may be also linked to digestibility since the apparent digestibility coefficient (ADC) of dry matter, which provides an overall indication of a diet’s digestibility, was significantly different among diets, as well as protein, lipid, and energy digestibilities (*p* < 0.01 in all cases, Fisher test) (Table 1). We observed that INS, YEA, and PAP diets had the highest ADCs in all cases, whereas SPI had the lowest ADC, thus describing the ability to convert and use feed energy into body weight. Similarly, in a previous study that included different species of cyanobacteria (*Spirulina maxima*) in tilapia feed, a strong increase in FCR and a decrease in digestibility was found. The amount of spirulina, however, was proportionally much higher (up to 100%) compared to ours (15%) [57]. The authors suggested that these observations may be due to the fact that spirulina was phosphorus limited [58]. Moreover, the VEG diet digestibility was also lower than that of INS, YEA, and PAP regarding dry matter and energy, which can be linked to a higher FCR for VEG diet. The digestibility decreases when increasing the amount of terrestrial vegetal protein was already documented in European sea bass [59]. This can be explained by the fact that plant-based ingredients contain antinutritive factors that reduce the digestibility of the diet as well as its palatability [60].

### 3.2. Changes of Protein Sources in Fish Diet do not Significantly Alter Gut Microbiota Diversity

The stability of gut microbial communities is an important factor affecting the overall health of marine fish [28]. To determine whether diet modification alters the diversity of gut microbiota, we performed a 16S rRNA sequencing analysis of the gut content in a subset of fish (*n* = 131) subjected to the six studied diets. Out of 1,396,548 sequence reads in all of the samples, we identified 1155 operational taxonomic units (OTUs at 97% of identity) representative of the sampled population as shown by rarefaction curves (Appendix A).

The OTUs richness of fish gut microbiota was not impacted by protein source modification, even after full replacement of fishmeal contained in the COM diet by full vegetal proteins in the VEG diet, and its derivates (Figure 4A). Similarly, Shannon’s diversity index, which accounts for both abundance and evenness of the species present, did not reveal any significant differences between the tested feeding regimens (Figure 4B). These results are in agreement with previous studies on the impact of fishmeal replacement on the α-diversity of intestinal microbiota of European sea bass [4,16], which reveals a high gut microbial adaptability to diet modifications.

### 3.3. Sea Bass Microbiota Composition Daisplays High Diet-independent Inter-individual Variability

Taxonomic analysis of bacterial sequences in gut samples identified a total of 22 different bacterial phyla, with 19 phyla in fish fed the VEG control diet, 19 with COM, 16 with INS, 20 with PAP, 17 with SPI, and 17 with YEA (Appendix A). Bacteria predominantly belonged to the phyla *Proteobacteria*, *Firmicutes*, *Actinobacteria,* and *Bacteroidetes* (Figure 5). These phyla usually represent up to 90% of the fish intestinal microbiota in different marine and freshwater species [28], including European sea bass [27,61,62] (Appendix A).

None of the OTUs identified were detected in all samples from animals fed with the different experimental feed diets, which is consistent with high inter-individual microbiota variability and shows that no “core microbiota” can be defined. In animals fed the control VEG diet, for example, this high variability was revealed at the genus level, although the microbiota was dominated by *Sphingomonas* sp. (relative abundance varying between 0.001% and 95.35%, detected in 53% of sampled individuals) and *Staphylococcus* sp. (relative abundance varying between 0.01% and 47.58%, present in 9/15 fish guts) (Appendix A). This high degree of inter-individual microbiota variability was already reported in sea bass [27,63], as well as in other fish species [64,65] and hypothesized to explain health and growth performance differences in aquaculture [65].

### 3.4. Diet does not Influence Microbiota Compositional Diversity

To determine how different gut bacterial communities are in fish fed with different experimental diets, we compared the taxonomic abundance profiles among fish gut samples of individuals fed different diets (β-diversity at OTU level) and used non-metric multidimensional scaling analysis (NMDS) based on Bray–Curtis dissimilarities. We showed that samples did not cluster according to the different diets (Figure 6A: Adonis test: *p* > 0.05), demonstrating that protein source variation does not influence sea bass intestinal microbiota under the tested conditions. When compared to the other sample types, all intestinal samples clustered together, indicating that the gut microbiota composition keeps its original features (Figure 6A). The observed inter-individual variability within diet groups is not explained by the different tanks in which fish were distributed, because all the intestinal samples clustered together, whatever aquarium they came from (Appendix A). Similarly, a previous study showed no significant difference in β-diversity of gut microbiota between fish fed with diets similar to the COM (20% FM/6% FO) and VEG diets used in our study. However, when intestinal mucosal microbiota communities were compared, significant differences were observed between fish fed FM and FO containing diet compared to terrestrial-plant-based diets [16]. These results suggest that microbial community is not only influenced by environmental factors such as water parameters and diet, but also by other intrinsic fish factors. We therefore cannot rule out that the absence of difference in ß-diversity observed in our analysis could be due to the type of sample used. Nevertheless, in general, the lack of a negative effect on microbial diversity and species richness is considered a desirable feature of sustainable diet replacements. A diet-induced reduction in commensal bacterial diversity or species richness may result in diminished colonization resistance against opportunistic pathogens which may enter the gastrointestinal tract of fish [66,67].

Sea bass are constantly exposed to microorganisms present in the water tank or the fish feeds, which could potentially influence the composition of fish microbiota [68]. We compared water, feed, and fish gut microbiota and showed that sea bass gut microbial composition is more similar to that of water than food, regardless of the used diets (Figure 6B, Adonis test: *p* > 0.05), suggesting that, although fish gut microbiota remains distinct from the microbial communities present in the water, water microbiota influences the gut bacterial population [31,69,70].

### 3.5. Growth does not Correlate with Diet-dependent Changes in Microbiota Composition

Differential abundance analyses of microbial taxa identified in sea bass microbiota in the different diet groups, relative to the VEG group, revealed a total of 17 differentially abundant families (Table 2). The majority of these families belong to the phylum *Proteobacteria* (59%), a predominant phylum in sea bass gut microbiota [27,61]. Eighteen families were significantly reduced in fish fed the experimental diets compared to fish fed the reference VEG diet (Table 2). Despite this, we did not detect any difference at the family level between the COM diet containing fishmeal and the full-vegetal VEG diet. At the genus level, the FM rich diet induced a significant decrease of relative abundance of *Lactobacillus* compared to the VEG-control diet (Table 3). A previous study comparing gut microbiota of sea bass fed FM and FO diet and a terrestrial plant-based diet showed no difference regarding the genus *Lactobacillus* [16]. The few differences observed between gut microbiota composition of fish fed the VEG and COM diets suggest that fishmeal replacement with vegetal terrestrial proteins does not have a major effect on intestinal microbial composition, and other elements should be considered, such as essential fatty acids, as previously described [16,27]. The high interindividual diversity observed within each diet group (Figure 5) suggests that a much higher number of animals would need to be analyzed to detect significant trends caused by diet interventions [27].

Diets displaying the best FCR values, i.e., INS, YEA, and PAP showed very different microbial alteration profiles. Fish fed INS showed 13 families with reduced relative abundance, compared to fish fed VEG (Table 2). This decrease in family abundance in INS-fed animals could be due to the fact that *Hermetia illucens*, the insect used in our study, is a rich source of small proteins and peptides that could have had balancing effects on specific taxa of the gut microbiota [71]. Furthermore, chitin-based insect exoskeleton is a form of insoluble fiber that has been suggested to display antimicrobial and bacteriostatic properties against several Gram-negative bacteria [72]. In contrast, fish fed the PAP diet, which also displayed an improved FCR value compared to VEG-fed animals, had the fewest number of significant changes in families when compared to the VEG diet, with only 3 families variation (Table 2). Among the altered genera, the major increase in relative abundance compared to the VEG diet concerned *Lactobacillus*, a genus that is also significantly more abundant with the VEG diet compared to COM-fed fish (Table 3). With the exception of *Lactobaciullus delbrueckii delbrueckii*, which positively affected fish growth [73], no effect on sea bass growth performance was established when different members of the genus *Lactobacillus* where exogenously included in fish diets [74,75,76]. However, members of the genus *Lactobacillus* are known to exert probiotic properties in aquaculture [77,78]; our results suggest that processed animal protein composed of poultry blood meal and hydrolyzed poultry feathers is associated with an increase of species belonging to the genus *Lactobacillus* which could thus have positive properties for the European sea bass. The YEA supplemented diet altered the abundance of 7 genera, including *Pseudomonas* (Table 3), four of which are reduced compared to VEG-fed animals. Yeast-derived glucans display immunostimulant properties in different freshwater and marine fish species [79], which could reduce the abundance of potential opportunistic pathogens belonging to the genus *Pseudomonas* [80].

By contrast, the SPI diet displayed the worst FCR value compared to the VEG control diet. However, animals fed this diet displayed microbial composition profiles similar to those fed other experimental diets with improved FCR values such as YEA (Table 2 and Table 3). These results could thus indicate that the differences observed in growth performance are not correlated to bacterial microbiota modifications.

## 4. Conclusions

In conclusion, our study showed that replacing only 15% of the vegetal feed by alternative protein from sustainable sources in fish feed formulation free of wild fishmeal has a significant impact on feed intake and fish performance. Feeds with positive effects could be produced with low-cost proteins (i.e., vegetal proteins) mixed with a minor proportion of high-cost proteins (i.e., insect or yeast proteins) to alleviate the detrimental effect of plant ingredients and thus improve feed intake and growth. This would provide opportunities to mix proteins from several origins to balance between performance and costs of the respective raw materials. Our results also showed that the different protein sources used for alternative fish diets were not associated to changes in microbial diversity and species richness but could induce alterations in the European sea bass gut microbiota. Interestingly, the intrinsic inter-individual variability of the European sea bass gut microbiota could contribute to its adaptability to protein source variation. Our study contributes a further understanding of the dynamic and complex relationships between diet, gut microbiota function and fish performance that will allow the development of aquafeeds for a sustainable fish aquaculture.

## Figures and Tables

**Figure 1 microorganisms-08-01346-f001:**
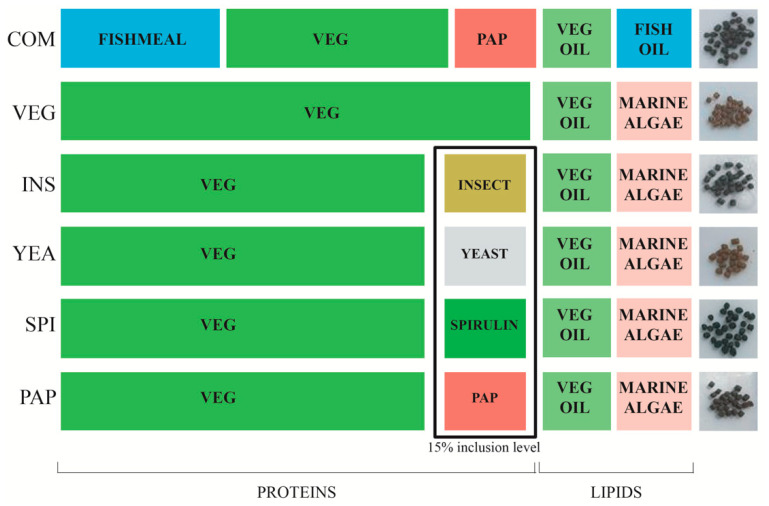
Profiles of fish feed formulations used in this study. The boxed ingredients show the experimental feeds in which 15% VEG feed were replaced by alternative proteins sources. COM: European commercial feed with conventional animal products. VEG: vegetal feed composed exclusively of terrestrial plant products and by-products. PAP: processed animal protein composed of a mix of poultry blood meal and hydrolyzed poultry feathers. INS: hydrolyzed insect meal (*Hermetia illucens*). YEA: yeast *Saccharomyces cerevisiae* meal. SPI: cyanobacterium (*Arthrospira platensis*). VEG oil: rapeseed oil. All diets, except COM, were enriched with monounsaturated fatty acids (MUFA) oil from marine micro algae (*Schizochytrium* sp). The images on the right show the color of experimental pellets.

**Figure 2 microorganisms-08-01346-f002:**
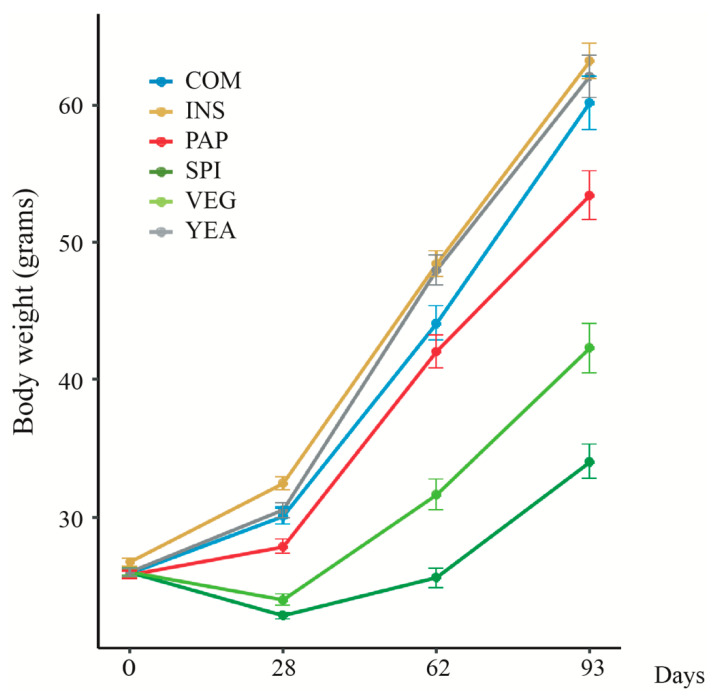
Evolution of tagged fish body weight gain according to used experimental diets. See Figure 1 for diet description. Body weights for each diet are presented as mean ± standard error (*n* = 159 for each diet except for COM with *n* = 53).

**Figure 3 microorganisms-08-01346-f003:**
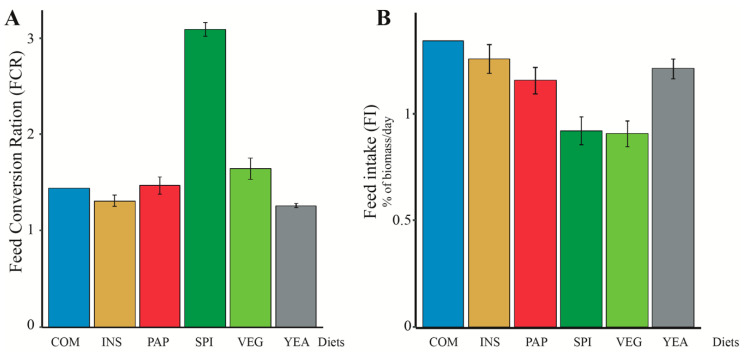
(**A**) Food Conversion Ratio (FCR) and (**B**) Feed Intake (FI) of fish according to used experimental diets. COM (conventional commercial feed control), INS (85% VEG + 15% insect), PAP (85% VEG + 15% poultry blood and feather), SPI (85% VEG + 15% cyanobacteria), VEG (100% terrestrial plants) and YEA (85% VEG + 15% yeast), presented as mean ± standard error (*n* = 3 for each diet except COM (*n* = 1)).

**Figure 4 microorganisms-08-01346-f004:**
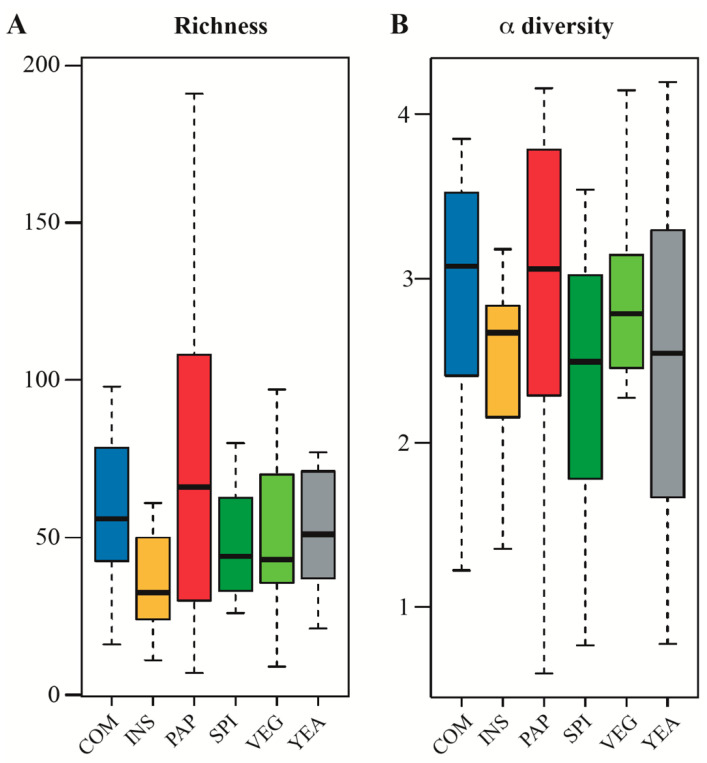
Measures of microbial α-diversity ((**A**): OTUs richness, and (**B**): Shannon index) in the gut of sea bass (*n* = 12 to 16 fish/diet).

**Figure 5 microorganisms-08-01346-f005:**
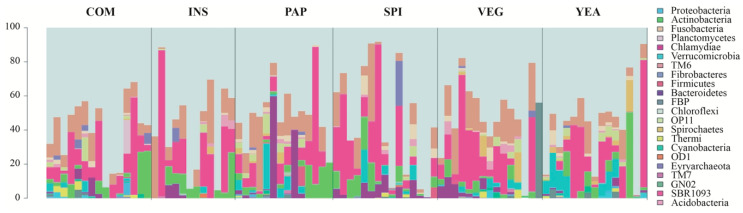
Relative abundances (%) of the 22 phyla identified in all samples of the different dietary groups shows a high variability of the gut microbiome among individual fish fed the different diets (*n* = 12 to 16 fish/diet). Top: fish feed formulations used in this study: COM (conventional commercial feed control), INS (85% VEG + 15% insect), PAP (85% VEG + 15% poultry blood and feather), SPI (85% VEG + 15% cyanobacteria), VEG (100% terrestrial plants), and YEA (85% VEG + 15% yeast).

**Figure 6 microorganisms-08-01346-f006:**
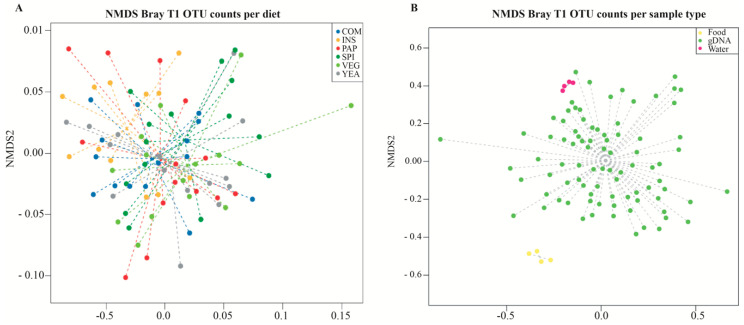
Non-metric multidimensional scaling plots based on Bray–Curtis dissimilarities showing the ordination between intestinal bacterial community of European sea bass fed the experimental diets (**A**) and sample type (**B**) at the end of the feeding trial (t = 93 days). Each dot represents an individual sample plot according to its microbial profile at OTU level (*n* = 12 to 16 fish/diet).Top right inset: fish feed formulations used in this study: COM (conventional commercial feed control), INS (85% VEG + 15% insect), PAP (85% VEG + 15% poultry blood and feather), SPI (85% VEG + 15% cyanobacteria), VEG (100% terrestrial plants), and YEA (85% VEG + 15% yeast).

**Table 1 microorganisms-08-01346-t001:** Apparent digestibility coefficient (ADC) of the different feeds tested according to the experimental diets. COM (commercial diet), VEG (100% terrestrial plants), INS (85% VEG + 15% INS), PAP (85% VEG + 15% poultry blood and feathers), SPI (85% VEG + 15% cyanobacteria), YEA (85% VEG + 15% yeast). Mean values are given ± standard deviation.

	COM *	VEG	INS	YEA	PAP	SPI
ADC dry matter (%)	81.9	70.4 ± 1.1 ^b^	76.9 ± 0.4 ^a^	73.2 ± 2.0 ^a,b^	72.5 ± 3.3 ^a,b^	68.2 ± 2.8 ^b^
ADC protein (%)	93.6	92.9 ± 0.9 ^a^	93.1 ± 0.4 ^a^	92.6 ± 0.9 ^a^	90.3 ± 2.6 ^a^	82.6 ± 1.7 ^b^
ADC lipid (%)	94.8	90.4 ± 0.5 ^a^	91.4 ± 0.4 ^a^	91.3 ± 0.3 ^a^	90.6 ± 2.6 ^a^	87.9 ± 1.5 ^b^
ADC energy (%)	89.4	81.1 ± 0.5 ^b,c^	85.2 ± 0.6 ^a^	84.6 ± 0.6 ^a,b^	84 ± 1.7 ^a,b^	79.7 ± 0.7 ^c^

Letters indicate significant difference between diets (Tukey test, *p* ˂ 0.05). * Fish feed formulations used in this study: COM (commercial feed control), VEG (100% terrestrial plants), INS (85% VEG + 15% insect), YEA (85% VEG + 15% yeast), PAP (85% VEG + 15% poultry blood and feather), and SPI (85% VEG + 15% cyanobacteria).

**Table 2 microorganisms-08-01346-t002:** Significantly different taxonomic groups at the family level. Significant log scaled fold-changes of differentially abundant families (*p*.adjust < 0.05) between experimental diets and the VEG control feed are presented (green: overabundant; orange: subabundant).

Phylum	Class	Order	Family	INS *	PAP	SPI	YEA	COM
*Proteobacteria*	*Alphaproteobacteria*	*Rhodospirillales*	*Rhodospirillaceae*	−8359	*-*	−8275	*-*	*-*
*Proteobacteria*	*Alphaproteobacteria*	*Rhizobiales*	*Hyphomicrobiaceae*	−8302	*-*	*-*	*-*	*-*
*Planctomycetes*	*Planctomycetia*	*Pirellulales*	*Pirellulaceae*	−5084	*-*	*-*	*-*	*-*
*Proteobacteria*	*Alphaproteobacteria*	*Rhizobiales*	*Phyllobacteriaceae*	−7244	−4738	−5618	−3665	*-*
*TM6*	*SJA-4*			−6680	*-*	−8252	*-*	*-*
*Proteobacteria*	*Gammaproteobacteria*	*Legionellales*	*Coxiellaceae*	−5239	*-*	−3614	−3442	*-*
*Proteobacteria*	*Gammaproteobacteria*	*Legionellales*		−5547	*-*	−3171	*-*	*-*
*Proteobacteria*	*Gammaproteobacteria*	*Legionellales*	*Legionellaceae*	−5194	*-*	*-*	*-*	*-*
*Firmicutes*	*Clostridia*	*Clostridiales*	*Clostridiaceae*	−5904	*-*	*-*	*-*	*-*
*OP11*	*WCHB1–64*	*d153*		−5139	*-*	*-*	*-*	*-*
*Bacteroidetes*	*Flavobacteriia*	*Flavobacteriales*	*Weeksellaceae*	−4257	*-*	*-*	*-*	*-*
*Bacteroidetes*	*Bacteroidia*	*Bacteroidales*	*Prevotellaceae*	−3516	*-*	*-*	*-*	*-*
*Proteobacteria*	*Alphaproteobacteria*			−3722	*-*	*-*	*-*	*-*
*Proteobacteria*	*Alphaproteobacteria*	*Rhizobiales*	*Methylobacteriaceae*	*-*	6551	7300	3892	*-*
*Proteobacteria*	*Betaproteobacteria*			*-*	5635	*-*	*-*	*-*
*Proteobacteria*	*Betaproteobacteria*	*Rhodocyclales*	*Rhodocyclaceae*	*-*	*-*	4820	*-*	*-*
*Proteobacteria*	*Alphaproteobacteria*	*Rhodobacterales*	*Rhodobacteraceae*	*-*	*-*	−3272	*-*	*-*
*Bacteroidetes*	*Chitinophagia*	*Chitinophagales*	*Chitinophagaceae*	*-*	*-*	3831	4628	*-*
*Actinobacteria*	*Actinobacteria*	*Actinomycetales*	*Micrococcaceae*	*-*	*-*	−3652	*-*	*-*
*Proteobacteria*	*Gammaproteobacteria*	*Pseudomonadales*	*Pseudomonadaceae*	*-*	*-*	*-*	−5510	*-*
*Firmicutes*	*Bacilli*	*Lactobacillales*	*Lactobacillaceae*	*-*	*-*	*-*	−4561	*-*
*Proteobacteria*	*Gammaproteobacteria*	*Pasteurellales*	*Pasteurellaceae*	*-*	*-*	*-*	−3922	*-*

* Fish feed formulations used in this study: INS (85% VEG + 15% insect), PAP (85% VEG + 15% poultry blood and feather), SPI (85% VEG + 15% cyanobacteria), YEA (85% VEG + 15% yeast) and COM (commercial feed control).

**Table 3 microorganisms-08-01346-t003:** Significantly different taxonomic groups at the genus level. Significant log scaled fold-changes of differentially abundant genera (*p*.adjust < 0.05) between experimental diets and the VEG control feed are presented (green: overabundant; orange: subabundant).

Phylum	Class	Order	Family	Genus	INS *	PAP	SPI	YEA	COM
*Proteobacteria*	*Alphaproteobacteria*	*Rhodospirillales*	*Rhodospirillaceae*		−8298	*-*	−8279	*-*	*-*
*Proteobacteria*	*Gammaproteobacteria*	*Legionellales*	*Legionellaceae*		−7147	*-*	−4141	*-*	*-*
*Proteobacteria*	*Alphaproteobacteria*	*Rhodobacterales*	*Rhodobacteraceae*	*Sulfitobacter*	−7449	*-*	−6105	−3795	*-*
*Proteobacteria*	*Alphaproteobacteria*	*Rhizobiales*	*Phyllobacteriaceae*		−6970	−4782	−5822	−3921	*-*
*TM6*	*SJA-4*				−6540	*-*	−8506	*-*	−4498
*Proteobacteria*	*Gammaproteobacteria*	*Legionellales*	*Coxiellaceae*		−4824	*-*	−3505	−3825	*-*
*Planctomycetes*	*Planctomycetia*	*Pirellulales*	*Pirellulaceae*		−4518	*-*	*-*	*-*	*-*
*Proteobacteria*	*Gammaproteobacteria*	*Legionellales*			−4592	*-*	*-*	*-*	*-*
*OP11*	*WCHB1–64*	*d153*			−5099	*-*	*-*	*-*	*-*
*Proteobacteria*	*Alphaproteobacteria*	*Rhodobacterales*	*Rhodobacteraceae*	*Marivita*	−5114	*-*	*-*	*-*	*-*
*Proteobacteria*	*Alphaproteobacteria*	*Rhodobacterales*	*Rhodobacteraceae*	*Paracoccus*	4100	4611	*-*	*-*	4942
*Verrucomicrobia*	*Verrucomicrobiae*	*Verrucomicrobiales*	*Verrucomicrobiaceae*	*Persicirhabdus*	*-*	3829	5959	5601	6226
*Proteobacteria*	*Alphaproteobacteria*	*Rhizobiales*	*Methylobacteriaceae*	*Methylobacterium*	*-*	4085	7628	3756	4152
*Firmicutes*	*Bacilli*	*Lactobacillales*	*Lactobacillaceae*	*Lactobacillus*	*-*	5097	*-*	*-*	−4518
*Proteobacteria*	*Gammaproteobacteria*	*Pseudomonadales*	*Moraxellaceae*	*Acinetobacter*	*-*	*-*	3217	*-*	2616
*Bacteroidetes*	*Chitinophagia*	*Chitinophagales*	*Chitinophagaceae*	*Sediminibacterium*	*-*	*-*	3647	4625	*-*
*Proteobacteria*	*Gammaproteobacteria*	*Pseudomonadales*	*Pseudomonadaceae*	*Pseudomonas*	*-*	*-*	*-*	−5295	*-*
*Proteobacteria*	*Gammaproteobacteria*	*Enterobacteriales*	*Enterobacteriaceae*		*-*	*-*	*-*	*-*	7736

* Fish feed formulations used in this study: INS (85% VEG + 15% insect), PAP (85% VEG + 15% poultry blood and feather), SPI (85% VEG + 15% cyanobacteria), YEA (85% VEG + 15% yeast) and COM (commercial feed control).

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
