# Peer review of "Growth Performance and Adaptability of European Sea Bass (Dicentrarchus labrax) Gut Microbiota to Alternative Diets Free of Fish Products"

_microorganisms, 2020, doi:10.3390/microorganisms8091346_

Round 1
Reviewer 1 Report
Overall the authors have presented important information regarding the intestinal microbial responses of European seabass to inclusion of vegetable and other protein sources. This information is novel and is worthy of pubication after some additional information has been added. I have a few reservations as the validity of the methodology used for sequencing of the microbiota samples, see comments below:
- Why did the authors use the V3-V4 region and not the V1-V2 region? Previously, we have used the V1-V2 region because it has been demonstrated that this region of the 16S rRNA gene is suitable and reliable to capture the diversity of the bacterial community from fish intestinal samples.
- The author's fail to present important information regarding the quality of their sequencing of the bacterial samples. There is no evidence of a good's coverage curve, indicating how much of the OTU's were captured by the sequencing. Moreover, there was no evidence of rarefraction curves to indicate whether the microbiota was adequately sampled and the sequencing depth was suffcient to analyse the bacterial diversity. I suggest the authors include this information to validate their sequencing approach.
Author Response
Please see a much clearer version of our responses in the uploaded PDF version with color and added figure
Overall the authors have presented important information regarding the intestinal microbial responses of European seabass to inclusion of vegetable and other protein sources. This information is novel and is worthy of pubication after some additional information has been added. I have a few reservations as the validity of the methodology used for sequencing of the microbiota samples, see comments below:
- Why did the authors use the V3-V4 region and not the V1-V2 region? Previously, we have used the V1-V2 region because it has been demonstrated that this region of the 16S rRNA gene is suitable and reliable to capture the diversity of the bacterial community from fish intestinal samples.
We understand the reviewer’s point. However, although V1-V2 region of 16S rRNA gene is commonly used to analyzed bacterial community diversity, the generated region is shorter with V1-V2 primers (<400 bp) than with V3-V4 primers (460 bp). This leads to a slight overestimation of species richness and use of V1-V2 region is less well adapted to the detection of minor community members than the use of larger region generated by V3-V4 primers.
In addition, most studies on sea bass gut microbiota composition used V3-V4 region of 16S rRNA gene and we therefore decided to use this region in order to be able to compare our results with published analyses of sea bass intestinal bacterial diversity.
- The author's fail to present important information regarding the quality of their sequencing of the bacterial samples. There is no evidence of a good's coverage curve, indicating how much of the OTU's were captured by the sequencing. Moreover, there was no evidence of rarefraction curves to indicate whether the microbiota was adequately sampled and the sequencing depth was suffcient to analyse the bacterial diversity. I suggest the authors include this information to validate their sequencing approach.
The reviewer has a good point. The quality of sequencing was assessed and verified but was not provided in the submitted manuscript. This information is now included in the result section of the revised manuscript and is associated to a new supplementary figure as follows:
Line 261-262: “Out of 1,396,548 sequence reads in all of the samples, we identified 1,155 Operational Taxonomic Units (OTUs at 97% of identity) representative of the sampled population as shown by rarefaction curves (Supplementary Figure S2).
Supplementary Figure S2: Rarefaction curves for each sea bass intestinal sample fed different diets. Rarefaction curves were assembled showing the number of operational taxonomic units defined at a 97% sequence similarity, relative to the number of total sequences. The vertical dotted-red line represents the plateau threshold for all curves (established to 10422 reads).

Reviewer 2 Report
This article is well written and interesting; before its publication, before its acceptance,however, some minor revisions are required:
line 17, could contribute to reduce the environmental impacts of fish farming, making on fishery industry more sustainable than in current state.
line 40, change footprint into impact
lines 42-43, change their impact into their effects on fish gut microbial community into microbiota, health and growth performance are not yet fully known;
line 42, after new protein sources please include a reference 7- Caruso (2015) Use of plant products as candidate fish meal substitutes: an emerging issue in aquaculture productions. Fisheries and Aquaculture Journal 6 (3), 1
line 115, initial biomass (B0) close brackets
line 197-202 seems a repetition of materials and methods (experimental diets composition)
In the paragraph Statistical analysis, only ANOVA referred to microbiota is reported, please include some indication of one-way ANOVA test used for comparison among the dietary treatments
In the captions to Figures 5 and 6 A and also in the table 1 , 2, 3 I suggest to explain in full the abbreviations used for the 6 diets
In fig. 5, the list of bacterial genera is hardly visible, please increase the character size
line 256, Changes of protein sources in fish diet DO not significantly alter
line 267, delete content
line 283, were detected in animals fed
line 294 change "between" into "among" fish gut samples of individuals fed different diets
line 308 delete that
In fig. 6, the legend is too small size to be clearly visible. Please enlarge it. In fig. 6B, the orange symbols (Water) are not visible, please check
line 444, animals in capital
line 476, Tenebrio molitor in italics
lines 529, 532 Dicentrarchus labrax in italics like at line522
reference 41 regarding total serum protein does not seem too much pertinent to the subject of this manuscript
last reference is not aligned
I suggest to update the references by including some more papers such as the following:
Rita Azeredo et al. 2017. The European seabass (Dicentrarchus labrax) innate immunity and gut
health are modulated by dietary plant-protein inclusion and prebiotic supplementation. Aquaculture
David Huyben et al. 2019 High-throughput sequencing of gut microbiota in rainbow trout (Oncorhynchus mykiss) fed larval and pre-pupae stages of black soldier fly (Hermetia illucens. Aquaculture
Mastoraki et al. 2020 A comparative study on the effect of fish meal substitution with three different insect meals on growth, body composition and metabolism of European sea bass (Dicentrarchus labrax L.) Aquaculture
Estruch G, Collado MC, Peñaranda DS, Tomás Vidal A, Jover Cerdá M, Pérez Martínez G, et al. (2015) Impact of Fishmeal Replacement in Diets for Gilthead Sea Bream (Sparus aurata) on the Gastrointestinal Microbiota Determined by Pyrosequencing the 16S rRNA Gene. PLoS ONE 10(8): e0136389. doi:10.1371/journal.pone.0136389
Author Response
Please see a much clearer version of our responses in the uploaded PDF version with color and added figure
This article is well written and interesting; before its publication, before its acceptance,however, some minor revisions are required:
- line 17, could contribute to reduce the environmental impacts of fish farming, making on fishery industry more sustainable than in current state.
We modified the sentence as follows:
Line 17: “Innovative fish diets made of terrestrial plants supplemented with sustainable protein sources free of fish-derived proteins could contribute to reduce the environmental impact of the farmed fish industry.
line 40, change footprint into impact
We agree and replaced footprint by impact Line 40.
- lines 42-43, change their impact into their effects on fish gut microbial community into microbiota, health and growth performance are not yet fully known;
We modified the sentence as follows:
Line 42-43: “While terrestrial plants, marine algae, yeast and alternative animal proteins, including insects, are currently considered as new protein sources, their impact on fish gut microbiota, intestine health, and growth performance are not yet fully known [7-10]”.
- line 42, after new protein sources please include a reference 7- Caruso (2015) Use of plant products as candidate fish meal substitutes: an emerging issue in aquaculture productions. Fisheries and Aquaculture Journal 6 (3), 1
We agree and included the Caruso reference.
7-Caruso G. Use of Plant Products as Candidate Fish Meal Substitutes: An Emerging Issue in Aquaculture Productions. Fisheries and Aquaculture Journal. 2015;6:1-3.
- line 115, initial biomass (B0) close brackets
Thank you. We corrected this typo and closed the brackets
Line 42-43: “The initial biomass (Bi) and final biomass (Bf)…»
- line 197-202 seems a repetition of materials and methods (experimental diets composition)
We understand the reviewer’s point. However, we consider that reminding the reader of the experimental design at the beginning of the result section was necessary to improve clarity and we therefore wish to keep this introductory sentence.
- In the paragraph Statistical analysis, only ANOVA referred to microbiota is reported, please include some indication of one-way ANOVA test used for comparison among the dietary treatments
We thank the reviewer for the comment. However, the different and specific statistical tests used are already indicated in the Material and Methods section of the manuscript:
Line 167-168: Description of the ANOVA test applied for microbiota comparison;
Line 178-180: Description of the ANCOVA test used for fish growth performance analysis
Line 184-186: Description of the ANOVA test applied to compare the feed intake, feed conversion ratio and digestibility of fish fed different diets.
- In the captions to Figures 5 and 6 A and also in the table 1 , 2, 3 I suggest to explain in full the abbreviations used for the 6 diets
We agree and we now also provide full explanation of the abbreviations used for the 6 diets is already included in the legends of Figure 5 and 6 as well as in table 1, 2 and 3
- In fig. 5, the list of bacterial genera is hardly visible, please increase the character size
We agree and increased the size of the characters used for bacterial genera (see below)
- line 256, Changes of protein sources in fish diet DO not significantly alter
Thank you. We corrected this typo
Line 256 “Changes of protein sources in fish diet do not significantly alter gut microbiota diversity. »
- line 267, delete content
We agree and deleted the word “content” Line 267
- line 283, were detected in animals fed
We wish to keep the original sentence (“were detected in all samples from animals fed with… ») because we want to underline the fact that it is not possible to define a core microbiota based on our results, precisely because none of identified OTUs were detected in all samples.
- line 294 change "between" into "among" fish gut samples of individuals fed different diets
We agree and changed the text accordingly.
Line 295-296-168: « …we compared the taxonomic abundance profiles among fish gut samples of individuals fed different diets (b-diversity at OTU level)… »
- line 308 delete that
We agree and changed the text accordingly.
- In fig. 6, the legend is too small size to be clearly visible. Please enlarge it. In fig. 6B, the orange symbols (Water) are not visible, please check
- We thank the reviewer for spotting this mistake: the legends corresponded to an earlier version of the figure. There are now only three items in the legends of Fig6 B, corresponding to the three colors. We also enlarged the legends and we expect that the production-ready high definition images (300 dpi) that will be uploaded at some point will be much clearer on print.
- line 444, animals in capital
We changed the text accordingly.
- line 476, Tenebrio molitor in italics
We changed the text accordingly.
- lines 529, 532 Dicentrarchus labrax in italics like at line522
We changed the text accordingly.
- reference 41 regarding total serum protein does not seem too much pertinent to the subject of this manuscript
We agree and removed this reference from the text.
- last reference is not aligned
We aligned the last reference
- I suggest to update the references by including some more papers such as the following:
Rita Azeredo et al. 2017. The European seabass (Dicentrarchus labrax) innate immunity and gut health are modulated by dietary plant-protein inclusion and prebiotic supplementation. Aquaculture
David Huyben et al. 2019 High-throughput sequencing of gut microbiota in rainbow trout (Oncorhynchus mykiss) fed larval and pre-pupae stages of black soldier fly (Hermetia illucens. Aquaculture
Mastoraki et al. 2020 A comparative study on the effect of fish meal substitution with three different insect meals on growth, body composition and metabolism of European sea bass (Dicentrarchus labrax L.) Aquaculture
Estruch G, Collado MC, Peñaranda DS, Tomás Vidal A, Jover Cerdá M, Pérez Martínez G, et al. (2015) Impact of Fishmeal Replacement in Diets for Gilthead Sea Bream (Sparus aurata) on the Gastrointestinal Microbiota Determined by Pyrosequencing the 16S rRNA Gene. PLoS ONE 10(8): e0136389. doi:10.1371/journal.pone.0136389
We thank the reviewer for these suggestions. However, we honestly do not see how adding these references to already fully referenced discussion points (corresponding to anterior studies) could usefully contribute to the manuscript, which does not constitute a review article and already contains 81 references total.
